# The Role and Therapeutic Implications of Inflammation in the Pathogenesis of Brain Arteriovenous Malformations

**DOI:** 10.3390/biomedicines11112876

**Published:** 2023-10-24

**Authors:** Ashley R. Ricciardelli, Ariadna Robledo, Jason E. Fish, Peter T. Kan, Tajie H. Harris, Joshua D. Wythe

**Affiliations:** 1Cardiovascular Research Institute, Baylor College of Medicine, Houston, TX 77030, USA; 2Department of Integrative Physiology, Baylor College of Medicine, Houston, TX 77030, USA; 3Department of Neurosurgery, Baylor College of Medicine, Houston, TX 77030, USA; 4Department of Neurosurgery, University of Texas Medical Branch, Galveston, TX 77555, USA; arrobled@utmb.edu (A.R.);; 5Toronto General Hospital Research Institute, University Health Network, Toronto, ON M5G 2C4, Canada; jason.fish@utoronto.ca; 6Laboratory Medicine & Pathobiology, University of Toronto, Toronto, ON M5S 1A8, Canada; 7Peter Munk Cardiac Centre, University Health Network, Toronto, ON M5G 2N2, Canada; 8Department of Neuroscience, University of Virginia School of Medicine, Charlottesville, VA 22903, USA; thh9t@virginia.edu; 9Brain, Immunology, and Glia (BIG) Center, University of Virginia School of Medicine, Charlottesville, VA 22903, USA; 10Department of Cell Biology, University of Virginia School of Medicine, Charlottesville, VA 22903, USA; 11Robert M. Berne Cardiovascular Research Center, University of Virginia School of Medicine, Charlottesville, VA 22903, USA

**Keywords:** brain arteriovenous malformation, cerebrovascular, vascular, endothelium, microglia, blood brain barrier, inflammation, neurovascular unit

## Abstract

Brain arteriovenous malformations (bAVMs) are focal vascular lesions composed of abnormal vascular channels without an intervening capillary network. As a result, high-pressure arterial blood shunts directly into the venous outflow system. These high-flow, low-resistance shunts are composed of dilated, tortuous, and fragile vessels, which are prone to rupture. BAVMs are a leading cause of hemorrhagic stroke in children and young adults. Current treatments for bAVMs are limited to surgery, embolization, and radiosurgery, although even these options are not viable for ~20% of AVM patients due to excessive risk. Critically, inflammation has been suggested to contribute to lesion progression. Here we summarize the current literature discussing the role of the immune system in bAVM pathogenesis and lesion progression, as well as the potential for targeting inflammation to prevent bAVM rupture and intracranial hemorrhage. We conclude by proposing that a dysfunctional endothelium, which harbors the somatic mutations that have been shown to give rise to sporadic bAVMs, may drive disease development and progression by altering the immune status of the brain.

## 1. Overview

Arteriovenous malformations (AVMs) are high-flow, low-resistance shunts that bypass capillary networks, resulting in the formation of a tangled, tortuous nidus [1]. According to the revised Richter-Suen classification, extracranial AVMs can consist of either a single dominant feeder artery (focal AVM), or multiple feeding arteries (multicentric AVM) [2,3], with extensive dilation of draining veins, and can lead to several complications such as ulceration, hemorrhage, and tissue necrosis, as well as congestive heart failure and left ventricular hypertrophy from decompensation [4]. Brain arteriovenous malformations (bAVMs), like their extracranial AVM counterparts, are characterized by shunting and the presence of a nidus [5] (Figure 1)**.** While they may share a similar pathophysiology to that of AVMs, bAVMs feature the unique risk of neurological deficits, including headaches, seizures, and intraparenchymal damage. Moreover, bAVMs pose a significant risk of intracranial hemorrhage [5], as they account for roughly 50% of all intracerebral hemorrhagic (ICH) strokes in the pediatric population [6] and around 5–6% of all ICH strokes in adults [7]. Previously unruptured bAVMs hemorrhage at a rate of 1–3% per year [8,9,10,11,12], with the risk of rupture rising to 5% after the first hemorrhage [13].

Herein, we discuss the epidemiology of these potentially deadly arteriovenous shunts, first summarizing the various inherited genetic syndromes that feature bAVMs, with a focus on hereditary hemorrhagic telangiectasia [14] and the second hit hypothesis, and then detailing the recently identified genetic variants in the RAS-MAPK signaling pathway underlying sporadic, non-familial bAVMs. Key findings from pre-clinical animal models for both genetic drivers of bAVM are briefly described, followed by an introduction to the immune landscape and cell types of the brain. Then, we explore the relationship between inflammation and bAVM pathogenesis, specifically focusing on vessel rupture and hemorrhage, and the potential of anti-inflammatory therapies in treating bAVM. Overall, this review addresses the etiology and pathogenesis of brain arteriovenous malformations from an endothelial-centric point of view, focusing on the interplay between endothelial dysfunction and immune cell dynamics and how their interplay impacts disease progression. 

## 2. The Etiology and Sequalae of Brain Arteriovenous Malformations

Patients who experience bAVM hemorrhage have an estimated mortality of 12–66.7% [15,16], with 23–40% of survivors having significant disability [17]. The asymptomatic prevalence of bAVMs is thought to be around 50 per 100,000 persons [18], with a detected prevalence of 10–18 per 100,000 and an incidence of 1.3 per 100,000 person-years [19]. 

Approximately 95% of bAVMs are unifocal (meaning one lesion per patient), non-familial (meaning there is an absence of family history for the disease) vascular anomalies [20]. Accordingly, these sporadic vascular anomalies are generally thought to arise from somatic—not germline—mutations. However, while genetic sequencing results of sporadic bAVM are relatively new (and discussed in detail below), much has been learned from studying less common causes of bAVMs, specifically inherited genetic syndromes that feature these devastating lesions such as hereditary hemorrhagic telangiectasia, or HHT.

HHT is characterized by epistaxis (nosebleeds), skin and mucosal telangiectasias (dilated postcapillary venule and arterial connections), as well as multiple AVMs with a preference for the lungs (30–50% of people with HHT) [21], liver (40–70%) [21,22,23], and brain (10%) [21,24]. There are two major subtypes of this autosomal dominant disorder, HHT1 and HHT2, which are each due to unique mutations impacting TGF-β signaling [25,26,27,28]. Specifically, HHT1, which accounts for 39–59% of all HHT diagnoses, is caused by a heterozygous loss of function mutations in the gene *ENDOGLIN*, which encodes for an essential type III accessory co-receptor for the TGF-β superfamily [29,30]. HHT2 involves loss-of-function variants in the gene *Activin A Receptor Like Type 1* (*ACVRL1*, *ALK1*), a type I TGF-β receptor [31]. While loss-of-function mutations in *ALK1* or *ENG* account for roughly 85% of all cases of HHT, approximately 5% of patients carry deleterious variants in *SMAD4,* which encodes an essential transcriptional effector of TGF-β signaling [32], while a handful of patients feature mutations within *BMP9*, which encodes the ligand for the ALK1–ENG receptor complex [33,34]. Mechanistically, the pathway can be briefly summarized as follows: circulating BMP9 and 10 bind to the type III co-receptor, Endoglin, and this ligand-receptor complex then binds transiently to ALK1. Endoglin is then released, and the type II BMP receptor, BMPR2, joins the BMP9/10–ALK1 complex, leading to BMPR2-depdent phosphorylation of ALK1, and activated ALK1 then phosphorylates the transcription factors SMAD1/5. Phosphorylated SMAD1/5 then bind SMAD4 and translocate to the nucleus to regulate endothelial gene expression. Simultaneously, phosphorylated SMAD1/5 block inactivation of PTEN, which prevents VEGF-mediated activation of PI3K/AKT signaling and endothelial proliferation and survival (for more details, please see [35]). Accordingly, loss-of-function mutations within *BMP9*, *ALK1*, *ENG*, and *SMAD4*, all lead to the increased endothelial dysfunction and proliferation characteristic of HHT. Given that HHT patients may feature bAVMs alongside arteriovenous malformations in other organs, and murine models of HHT phenocopy these multi-organ vascular anomalies, studies on this disease have yielded extensive insights into bAVM pathophysiology (for an excellent review on this topic, please see PMID: [36].

In HHT patients, the tissues most prone to developing telangiectasias are those subjected to repetitive injury, damage, inflammation and repair, such as the face, lips, and fingers [21]. Importantly, studies in animal models (which are beyond the scope of this review) are consistent with these findings. Collectively, these observations suggested that a second “hit”, such as the presence of a novel variant in a modifier gene, or a pro-inflammatory/angiogenic cue, together with reduced TGF-β signaling resulting from a germline loss-of-function mutation, lead to AVM formation [37,38]. Recent next-generation sequencing studies suggest the second hit may also be the acquisition of an additional somatic loss-of-function mutation in a HHT-related gene [39].

Our understanding of these sporadic vascular anomalies changed dramatically in 2018 with the discovery from our own group [40], and validation soon after by others [41,42,43,44,45,46], that >50% of sporadic bAVM cases contain gain-of-function genetic variants within a gene encoding a small GTPase, *KRAS*. RAS proteins, which cycle between an inactive GDP-bound state and an active GTP-bound state, typically function downstream of receptor tyrosine kinase signaling, and can activate numerous effector pathways, ranging from the RAF/MEK/ERK network to PI3K/AKT signaling. Since that seminal discovery of *KRAS* mutations, other studies have identified mutations within the downstream serine/threonine kinase *BRAF* [41,43,44], and activating *MAP2K1* mutations within extracranial AVMs [47].

Additionally, whole-exome sequencing studies have found various mutations in the BMP/TGF-β pathway, including *MAP4K4*, which encodes a kinase responsible for the phosphorylation of SMAD1, *ZFYVE16*, which encodes a SMAD1 anchor important for both the phosphorylation and formation of the SMAD2–SMAD4 complex, and LEM3, which interacts with SMAD1/5/8 to antagonize BMP signaling [48]. Wang and colleagues further identified potentially pathologic variants in VEGF-VEGFR2 signaling in patients with bAVM. Missense variants were identified in *SCUBE2* (c.1592G>A (p.Cys531Tyr)), which enhances VEGF/VEGFR2 binding, *TIMP3* (c.311T>C (p.Leu104Pro)), which blocks VEGF/VEGFR binding, and *SARS* (c.971T>C (p.Ile324Thr)), which inhibits *vegfa* transcription [49,50], while a truncating variant was identified in *PITPMN3* (c.274C>T (p.Arg92Ter)), which encodes a transmembrane receptor for CCL18 and promotes PI3K/AKT-dependent cell migration (and PI3K/AKT signaling is also activated by VEGF) [48,51]. A follow-up exome sequencing study comparing the genome of 112 affected Han Chinese patients to that of their two unaffected parents (i.e., a trio sequencing study) identified 16 genes with compound heterozygous mutations, 2 of which were mutated in three trios: *LRP2*, a multi-ligand endocytic receptor and receptor for the pro-inflammatory molecule lipocalin 2, and *MUC5B*, which encodes respiratory tract mucin glycoproteins, a puzzling finding due to its lack of established role in angiogenesis and blood vessel function [52]. Five of the remaining genes, found in two trios, had notable roles in angiogenesis and vessel stability: *MYLK*, which is implicated in heritable thoracic aneurysms and spinal arteriovenous malformations [53,54], *PEAK1*, which is involved in VEGFR2 expression [55], *PIEZO1*, which acts as a mechano-transducer with several endothelial functions [56], *HSPG2*, which produces the protein perlecan, which maintains extracellular membrane integrity [57], and *PRUNE2*, which is associated with aneurysmal tissue [58]. The treatment of symptomatic bAVMs, whether as a part of a complex inherited genetic syndrome (such as HHT), or as an isolated sporadic lesion initiated by a somatic variant within the endothelium (e.g., *KRAS* gain of function), represents an incredible challenge for neurosurgical teams, due to the fragility of intracranial vessels and the associated risk of stroke and hemorrhage [59]. The decision for intervention is patient-specific and depends on bAVM eloquence, size, location, vascular anatomy, and complexity, as well as hemorrhage history [60]. 

Conventional treatments include open surgical resection, stereotactic radiosurgery, and endovascular embolization, either alone or in conjunction with another modality [61,62]. Of these interventional options, open surgical resection continues to be recognized as the gold standard for treating AVMs, especially for low-grade lesions, owing to its superior efficacy, safety characteristics, high rates of complete obliteration, and rapid results, mitigating hemorrhage risk [63]. Endovascular embolization is typically used as an adjunct, either pre-surgical embolization, or before or after radiosurgery, with the possibility of achieving a cure in a small percentage of AVMs [62]. Stereotactic radiosurgery offers a minimally invasive technique, that is most appropriate for small- or medium-sized AVMs in deep or eloquent brain regions [64]. For younger patients with optimally chosen, small-volume lesions, stereotactic radiosurgery obliteration rates have reached up to 80%. However, the full range of beneficial and adverse impacts has a latency period and may not become fully evident until years later [65]. Furthermore, the definitive establishment of the safety and efficacy of stereotactic radiosurgery is not yet conclusive, with reduced radiographic complete obliteration rates, especially for large-volume bAVMs, ruptured bAVMs, cases post-embolization, and bAVMs with a higher Spetzler–Martin grade [66,67,68,69].

Currently no FDA-approved drug therapies exist for treating bAVM, although efforts are focused on the use of FDA-approved MEK1/2 inhibitors, as the MAPK kinase cascade (RAF-MEK-ERK) is an obligate target of KRAS in the endothelium [40,70,71,72,73]. Additionally, given the recent demonstration that endoluminal biopsy of a bAVM can be followed by exome sequencing [74], if lesions genotyped as positive for the G12C- or G12D-activating variants in *KRAS*, specific inhibitors targeting these variants have been approved for treating various cancers and could be of potential use in these cases [75,76]. Whether either of these agents will successfully normalize vascular malformations in patients, or at the very least prevent lesion progression, remains to be determined. Furthermore, if a positive response is seen, whether patients will later develop therapeutic resistance to these drugs, as has been shown for both MEKi and BRAF inhibitors [77,78], and perhaps now KRAS-specific inhibitors in cancer [79,80], is also unknown. Others have suggested that AVMs may be sensitive to anti-angiogenic therapies in the context of HHT [14,81,82,83], although clinical trials are needed to validate the efficacy of this approach. 

With limited treatment option for these patients, there is an urgent need to identify novel therapeutic targets based on our current understanding of bAVM pathogenesis. Since the single greatest risk bAVM patients face is the possibility of lesion rupture and intracerebral hemorrhage, identifying factors that promote or prevent rupture is critical for understanding lesion progression and identifying novel potential therapies that may reduce the risk of intracranial hemorrhage. Inflammation, in particular, proves an important target for drug research based on both the “second hit” model and its overall association with intracranial hemorrhage in both bAVMs and other cerebrovascular conditions [84]. Thus, we must first examine inflammation and the cell signaling networks controlling this process, and how it impacts blood–brain barrier (BBB) integrity and the neurovascular unit (NVU) in the setting of bAVM.

## 3. Inflammation and the Neurovascular Unit

The near-impenetrable nature of the BBB, combined with the assumed absence of a lymphatic conduit in the brain parenchyma for trafficking brain-derived antigens and immune cells, led many to conclude that the brain was an immune-privileged tissue [85]. This view has been amended due to recent data showing that the meningeal borders of the CNS are under continuous immune surveillance [86,87,88] and that the outermost layer of the three meningeal membranes that envelop the central nervous system (CNS), the dura mater, contains lymphatic vessels that drain to cervical lymph nodes, which eventually return fluid to the venous circulation [89,90]. Recent studies demonstrated that these lymphatic vessels not only drain excess fluid, antigens, and waste from the cerebral spinal fluid (CSF) [90,91,92,93], but they also provide a conduit for immune cell trafficking to and from the CNS [94,95,96,97]. Thus, while the BBB limits many pathogens and cells from traversing through the circulation into the underlying parenchyma, the healthy brain is not an immunological desert. 

Within the brain, the neurovascular unit (NVU), which is the subunit of the BBB, consists of endothelial cells, vascular smooth muscle cells, pericytes, microglia, and glial progeny (astrocytes, neurons, and oligodendrocytes) [98] (Figure 2). The orchestrated response of the myeloid compartment, including microglia in the parenchyma, dendritic cells in the meninges (dura, arachnoid, and pia), choroid plexus, and perivascular spaces, perivascular macrophages along the cerebral blood vessels in the choroid plexus, recruited monocytes from the bone marrow, and various immune cells in the leptomeninges [86,88,97,99], all combine to generate a robust protective and restorative response to insult or injury in the brain. The metabolic demands of these cells within the NVU and the rest of the brain are remarkably disproportionate compared to those within the rest of the body, as the brain utilizes upwards of 20% of all available oxygen and glucose, while only accounting for roughly 2% of the overall body mass [100]. Accordingly, the NVU and BBB, and the resultant delivery of oxygen and nutrients to the underlying brain parenchyma by cerebral blood flow that they enable, are essential for baseline neurological function. While NVU dysfunction and the impact of reducing cerebral blood flow and thus limiting the supply of available oxygen and nutrients to the brain are unquestionably detrimental, there is also a concomitant reduction in the clearance of neurotoxic substances such as β-amyloid and α-synuclein, and both of these decreased functions can contribute to numerous cerebrovascular pathologies, including stroke [101,102,103] and leukodystrophies [104], as well as neurodegenerative/neuroinflammatory disorders, such as Alzheimer’s, Parkinson’s and amyotrophic lateral sclerosis [105]. 

Notably, BBB breakdown and vascular dysfunction (e.g., cerebral hypoperfusion) can precede cognitive decline and clinical symptoms in numerous neurological pathologies [106], including aging, multiple sclerosis [107,108,109], stroke, Alzheimer’s [110,111,112,113,114], and epilepsy, with the disruption of vascular function in the CNS causing ion disturbance, edema, and neuroinflammation [115]. These observations have prompted researchers to investigate if either BBB dysfunction contributes to, or drives, lesion rupture in bAVM rupture as well. Specifically, recent scRNA-sequencing studies suggest that compromised BBB function, inferred from loss of mature CNS endothelial BBB markers and increased expression of peripheral endothelial genes, could lead to an influx of inflammatory cells that may drive bAVM rupture [116]. Additionally, *PLVAP*, a marker of fenestrated endothelium that is typically expressed by angiogenic, peripheral endothelium (e.g., outside of the mature CNS), and *ANGPT2*, an inducer of EC permeability, are upregulated in ECs in the nidus of bAVMs, suggesting that endothelial barrier function may be compromised in these lesions [116,117]. Furthermore, transcriptomic analysis revealed an upregulation of genes characteristic of developing fetal vasculature, implying that the vessels in the nidus are either immature and/or actively remodeling [116]. Gene ontology analysis of differentially expressed genes in the endothelium of the nidus highlighted angiogenic, inflammatory, and endothelial-to-mesenchymal transition pathways, while the computational prediction of ligand–receptor interactions highlighted EC–pericyte and EC–immune cell interactions associated with inflammation [116,117]. Below, we will explore this concept in more depth by first defining and analyzing various inflammatory cells and their role in healthy tissues and then describing their potential involvement in bAVM rupture and pathogenesis. 

**Figure 2 biomedicines-11-02876-f002:**
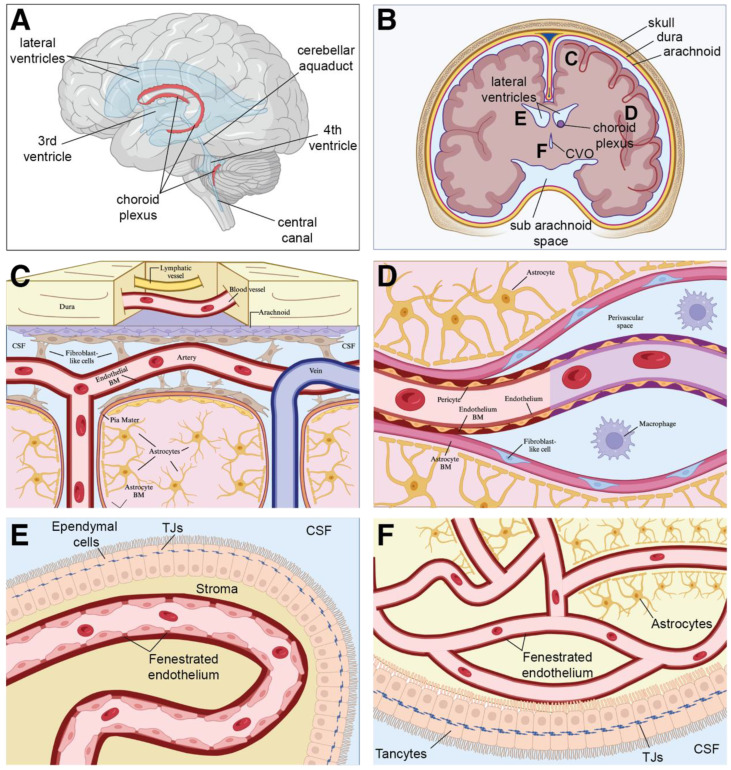
Cerebrovascular endothelium and lymphatics in the adult brain. (**A**) Schematic, sagittal view of the human brain (**B**) Coronal view of the brain, with the major middle cerebral arteries indicated in red. (**C**) Meningeal and cortical vascular networks within the brain. Within the dura, lymphatic blood vessels and fenestrated blood vessels lacking tight junctions allow for the transport of cells and molecules. The adjacent arachnoid, an epithelial layer, provides a barrier between the peripheral vasculature of the dura mater and the cerebral spinal fluid (CSF) via the presence of efflux pumps, channels, and tight junctions. Within the pia, leptomeningeal blood vessels, devoid of astrocytic ensheathment, are connected through tight junctions. Penetrating pial arteries located below that dura–arachnoid interface perfuse the underlying parenchymal tissue and are covered by a densely packed perivascular layer of astrocytic end feet and their processes, astrocytic (yellow), pial (pink), and endothelial (red/blue) basement membranes; and smooth muscle cells. (**D**) A capillary venule surrounded by the capillary endothelial basement membrane (purple) and the astrocytic basement membrane (pink). In contrast, the post-capillary venule contains a CSF fluid-filled perivascular space between the two membranes. Fibroblast-like cells (blue) line the astrocytic basement membrane of the post-capillary venule, serving as an extension of the pia mater. (**E**) The blood–CSF barrier is formed by tight joints in the choroid plexus. Within the choroid plexus, vessels are fenestrated to allow for molecular exchange. (**F**) The vasculature surrounding the perimeter of circumventricular organs is surrounded by a BBB with the astrocytic foot process, similar to vessels within CNS parenchyma. While the vessels themselves are fenestrated, ependymal tancycytes surround CVOs and contain tight junctions preventing CSF movement. Abbreviations: CVO = circumventricular organ, TJs = tight junctions, and CSF = cerebral spinal fluid. Figure adapted from [118].

## 4. Immune Cell Populations within the Brain

### 4.1. Resident Immune Cells in the Brain—Microglia

Macrophages act as sentinels for foreign pathogens and local injury cues, secreting both pro-inflammatory and pro-angiogenic factors such as TNF-α, IL-6, VEGF, and MMP-9, and also regulating the phagocytosis of pathogens and debris [119,120]. Macrophages express both cell surface and endosomal pattern recognition receptors (PRRs), including toll-like receptors (TLRs). In response to either pathogenic signals (PAMPs) or endogenous damage-associated signals (DAMPs), PRR signaling activates downstream pathways, such as the NF-κB signaling cascade, to stimulate cytokine production or major histocompatibility complex (MHC) expression [121,122,123,124]. Once TLRs, or other similar scavenger receptors (e.g., nod-like receptors; C-type lectin receptors) are bound by their appropriate ligand, they also stimulate T cell activation, triggering the adaptive immune system response [125].

While the majority of macrophages in peripheral tissues originate in the bone marrow, microglia in the brain derive from embryonic yolk-sac progenitors around E10.5. These fetal-derived cells persist and maintain the microglial pool within the adult brain in both mice and rats [126,127,128,129]. In the absence of pathology, microglia are the sole immune cell type within the brain parenchyma (Figure 2). While they are found within both white and grey matter, their distribution varies considerably across different regions of the brain. Microglia density is greatest within the hypothalamus and neostriatum, and lowest in the cerebellum, medulla oblongata, and spinal cord [130,131,132,133,134]. In addition to these regional differences, recent studies show that microglia feature a remarkable degree of heterogeneity in terms of cell morphology, motility, gene expression signature, and ultimately function, as these gardeners and sentries of the CNS are far from being a homogenous population [133]. Finally, these cells contribute to a myriad of processes within the CNS, ranging from neuronal circuit development to neurogenesis [135,136,137,138,139]. 

In response to many pathologic insults in the brain, such as ischemic stroke or neurodegenerative disease, microglia activate, changing from a highly ramified morphology into an amoeboid shape. Microglial activation increases their phagocytic activity and cytokine expression, which leads to the subsequent recruitment of peripheral immune cells to the CNS [140]. Furthermore, microglial activation regulates synaptic pruning [141]. Dysregulated synaptic pruning by microglia may contribute to both neuropsychiatric disorders and neurodevelopmental degenerative diseases (NDD) [142,143]. However, this developmental role is contentious, as compromised microglia development in mice does not lead to overt signs of NDD or altered CNS phenotypes [144,145]. Some propose that these paradoxical data are explained by astrocyte compensation, as they can expand their phagocytic territory, as well as other functions, to compensate for microglia loss in order to maintain synaptic homeostasis [146,147]. 

Rather than acting in isolation, microglia often cooperate with other members of the NVU, such as endothelial cells, to promote pro-inflammatory responses in the brain [148]. When activated, microglia can use both cytokines and chemokines to induce EC expression of adhesion molecules, allowing for lymphocytic infiltration [149]. The subsequent neuroinflammation can impair BBB and vascular function [148,150]. While microglia can communicate with the endothelium, endothelial cells, too, can impact microglia, stimulating their release of inflammatory cytokines and the production of reactive oxygen species [151]. While the identity of the cell type that initiates the inflammatory state is not known, it is nonetheless clear that microglia and endothelial cell cross-talk is important in neuroinflammation [148]. Physical contact of microglia with endothelial cells can lead to downregulation of endothelial cell tight junction and adherens junction components, such as VE-cadherin, Occludin, and Claudin-5 [150,152]. Notably, activated microglial also secrete IL-1β and TNF-α, which decrease expression of genes encoding tight junction proteins in the endothelium, further contributing to BBB permeability [153,154]. 

Recently, expansive single-cell gene expression profiling of tissue from mouse models of Alzheimer’s disease, amyotrophic lateral sclerosis (ALS), and aging, identified a novel subset of microglia: disease-associated migroglia (DAMs) [155]. DAMs feature a unique transcriptional and functional signature, retaining typical microglia markers (e.g., *Iba1*, *Cst3*, and *Hexb*) while downregulating expression of homeostatic genes (*P2ry12*, *P2ry13*, *Cx3cr1*, *CD33*, *Tmem119*, etc.) and upregulating pathways involved in lysosomal, phagocytic, and lipid metabolism, as well as several known Alzheimer’s disease risk factor genes (e.g., *Apoe*, *Ctsd*, *Lpl*, *Tyrobp*, and *Trem2*) [146]. Critically, studies in numerous murine models of neurodegenerative pathogenesis, from Alzheimer’s to ALS, show that DAMs localize to regions in the CNS affected by each disease [146]. These same markers are present in human AD post-mortem brains [155,156], suggesting that irrespective of the underlying disease etiology, the microglial transition to a DAM phenotype is a common response in CNS pathogenesis. Similar to the recognition of PAMPs and DAMPs by pattern recognition receptors in the innate immune response, neurodegeneration-associated molecular patterns (NAMPs), signals such as myelin debris, protein aggregates (Aβ plaques, tau tangles), and neuronal apoptotic bodies, are recognized by cell surface receptors enriched on DAMs, which then act to contain and remove the damage. Induction of the DAM phenotype is only one piece of an increasingly complex puzzle that is neurodegenerative disease, as alteration in the peripheral immune system, dysregulation of other cell types in the NVU, and particularly loss of blood–brain barrier integrity, all contribute to the onset and progression of neurodegenerative disease. Whether DAMs, or even microglia, interact with the cerebrovasculature in the setting of vascular malformations is unknown, but these immune cells may very well contribute to bAVM progression and/or rupture and thus may represent attractive therapeutic targets in the future.

### 4.2. Resident Cells in the Brain—Astrocytes

Another resident cell type within the brain and a major component of the NVU, astrocytes, also play a key role in neuroinflammation. Like microglia, astrocytes are also important for immune cell activation and trafficking. Activated astrocytes are also associated with pro-MMP secretion and high levels of MMP9 [157]. Depending on the context, astrocytes also release anti-inflammatory cytokines that induce tissue repair [158,159,160]. Furthermore, astrocytes can activate microglia, promoting an anti-inflammatory phenotype by directing microglia to engulf debris and suppress the inflammatory response [151]. Microglia have also been implicated in the activation of astrocytes [161], thus the interactions between these cell types may be involved in the immune response to bAVMs.

### 4.3. Resident Cells in the Brain—Pericytes

In addition to their endothelial lining, blood vessels also contain mural cells that associate with and ensheathe lumenized endothelial tubes. At a simplistic level, depending on their location, density, morphology, and gene expression profile, mural cells can be divided into one of two categories: vascular smooth muscle cells or pericytes. Whereas vascular smooth muscle cells wrap around large diameter vessels, such as arteries and veins, in multiple concentric layers, pericytes associate with smaller diameter vessels such as arterioles, capillaries, and venules [162]. Pericytes, situated between endothelial cells and astrocytic foot processes, share the basement membrane of the CNS endothelium, and ensure vessel stability and homeostasis (Figure 3). Pericytes can feature one of three distinct morphologies, depending on their location and shape: mural pericytes, which are located at the transition between arterioles and capillaries, mesh pericytes, confined to the first and second branches of post-arteriole capillaries, and thin-strand pericytes, localized to the second branch of post-arteriole capillaries [163]. Like other mural cell types, such as smooth muscle cells, pericytes can be found throughout the body; however, capillary vessels within the CNS have been found to have the greatest ratio of pericytes to endothelial cells [164,165]. Pericytes, which envelop the endothelium, are a central component of the NVU, and as such are key mediators of BBB integrity [166,167,168]. These cells are thought to modulate cell signaling networks that regulate endothelial permeability in the brain [166,167] and also control polarization of the astrocytic foot processes surrounding CNS vessels [167]. Accordingly, they regulate BBB permeability, angiogenesis, clearance of toxic metabolites, capillary hemodynamic responses, and even neuronal stem cell activity [168]. In addition to these functions, pericytes also play an important role in neuroinflammation by polarizing toward one of two states: a neuroprotective state and a proinflammatory state [163]. The neuroprotective state stimulates both endothelial [169] and neuronal [170] cell survival, while their proinflammatory state enhances the inflammatory response by producing factors such as MMP-9, nitric oxide, chemokines, and chemoattractant cues that act to promote monocytes and macrophage recruitment [163]. Cross-talk and regulation of these two states is paramount for mitigating damage in vascular diseases such as cerebral ischemia, as pericytes stabilize disrupted endothelial cells and regulate expression of inflammatory factors (e.g., TNF-α, IFN-γ, IL-1, IL-6, and IL-12) to promote pericyte migration and proliferation and act to regulate immune cell trafficking into the CNS [163]. 

### 4.4. Peripheral Immune Cells in the Brain

Peripheral immune cells such as monocytes, neutrophils, B and T cells are recruited to the brain in many pathologic conditions, and help permeabilize the BBB, and contribute to the inflammatory process together with microglia [171]. The endothelium plays a role in peripheral immune cell recruitment [172], with important steps highlighted in Figure 4. Each lineage will be briefly discussed below.

### 4.5. Monocytes

Monocytes, a population of leukocytes (e.g., white blood cells) conserved across vertebrates, are mononuclear phagocytes with critical, but distinct roles in tissue homeostasis and immunity that protect the body against infectious disease and foreign pathogens [173,174]. Put simply, monocytes can be thought of as key mediators of inflammation during pathogen challenge. These bone marrow-derived leukocytes circulate in the blood and spleen, traffic to organs to differentiate into macrophages, respond to danger signals via pattern recognition receptors, and are known for their ability to phagocytose debris, produce cytokines, present antigens to lymphocytes, regulate inflammation, and promote angiogenesis [174]. Monocytes fall into one of three categories: classical, non-classical, and intermediate [175,176,177]. 

Arising from myeloid precursor cells during embryonic and adult hematopoiesis, classical inflammatory monocytes (CD14^+^CD16^−^CCR2^+^ in humans and Ly6C^high^CCR2^+^ in mice) exhibit a remarkably short half-life; approximately 20 h in mice [178,179]. They proliferate within their primary lymphoid organ niche (e.g., the bone marrow bone or fetal liver) in response to an exogenous insult, such as injury or infection, and subsequently extravasate into the circulation in a CCR2-dependent mechanism. They then home to the site of inflammation using adhesion molecules and chemokines receptors, where they will then carry out key immune functions [180]. They also secrete a distinct repertoire of chemokines that will, in turn, recruit other immune cells, while also presenting antigens via class I and class II MHC molecules [181]. Critically, like the bone marrow, the spleen may act as a monocyte reservoir in mice, and these cells can be mobilized in response to signals emanating from a distant tissue [182]. In addition, monocytes may also replenish depleted pools of tissue-resident macrophages [183]. However, microglia in the brain originate, and are maintained, independent of input from recruited monocyte cells [179,184,185].

Non-classical patrolling monocytes (Ly6C^low^CX3CR1^high^ in mice and CD14^dim^CD16^+^CX3CR1^high^) crawl along the intraluminal surface of vessels, surveying vascular integrity and phagocytose-injured endothelial cells and debris, while also recruiting neutrophils to injury sites [176,186]. They can also differentiate into dendritic cells [187,188,189]. Reports conflict as to whether these cells secrete pro-inflammatory cytokines [176,190,191] or act in an anti-inflammatory manner to resolve inflammation and promote healing and fibrosis [176,192,193]. These non-classical monocytes continuously patrol the luminal side of the vascular endothelium in both homeostatic and inflammatory conditions, and have been directly observed in veins and arteries of the brain [194].

Intermediate monocytes (Ly6C^int^CX3CR1^high^ in mice; CD14^+^CD16^+^CXCR1^high^ in humans) have been identified, and while their function may be distinct from classical and non-classical myocytes, they have been incompletely characterized [177]. While the exact role of the intermediate monocyte has yet to be elucidated, they have been shown to both increase in inflammatory and chronic conditions such as cardiovascular disease, rheumatoid arthritis, and Crohn’s disease, and produce pro-inflammatory mediators in response to bacteria, expand during infection, and present antigens [195,196,197,198,199]. Under steady-state conditions, elegant Cre-*loxP* based fate mapping demonstrated that monocytes, or other bone marrow-derived populations, do not contribute to the maintenance of macrophage populations in peripheral tissues, including the brain, in adults [179,184]. 

### 4.6. Dendritic Cells

Found in the mammalian choroid plexus and meninges [200,201,202,203], multiple subsets, namely conventional DCs, plasmacytoid DCs, and inflammatory DCs, have been reported [204]. With their nomenclature and phenotypic markers only recently standardized [205,206], their definitive role under homeostatic conditions remains unclear. Immunologically, the main functions of DC’s are T cell activation via antigen presentation as well as cytokine release [207]. From an inflammatory standpoint, DCs are critical in initiating the immune response as well as promoting self-tolerance [208]. 

### 4.7. Neutrophils

Neutrophils are responsible for helping initiate the immune response and are often the first responder of WBCs that enter damaged or stressed tissue [209]. Upon encountering a microbe, they exhibit three main functions: phagocytosis, degranulation, and microbial trap formation via NETs (neutrophil extracellular traps) [210]. NETs are particularly important for antimicrobial activity as their chromatin fiber meshworks with granule-derived antimicrobial peptides and enzymes allow for pathogens to be immobilized and subsequently killed [211]. While originally known for these antimicrobial functions, studies have found that they also play an important role in cellular cross-talk, aiding with the regulation of DCs, T, and B cells, NK cells, and macrophages [210]. Additionally, recent studies have highlighted the role of neutrophils in the initiation and resolution of inflammation through the production of cytokines and inflammatory factors [212,213]. One such inflammatory factor, MMP-9, has been found to largely stem from neutrophils that both produce the factor as well as prevent its degradation through the release of NGAL, neutrophil gelatinase-associated lipocalin, which complexes with MMP-9 [214,215]. 

### 4.8. T and B Cells

T and B cells are derived from hematopoietic stem cells (HSCs) in the fetal liver, and, later, the bone marrow. To become T cells, HSCs differentiate into multipotent progenitors, after which subsets transcribe recombinant activating gene one and two (RAG1 and RAG2) to become lymphoid-primed multipotent progenitors and then common lymphoid progenitors. These progenitors are then able to migrate to the thymus (early thyroid progenitors) to become mature T cells [216]. B cells undergo a similar process, with HSCs activating transcription factors and differentiating to early lymphoid progenitors and then common lymphoid progenitors. Common lymphoid progenitors then can differentiate into DCs, NKs, or common lymphoid progenitor (CLP-2) cells. CLP-2 cells then travel to lymph nodes to mature into functional B cells [217,218,219]. 

T cells consists of two main subtypes, CD4^+^ T cells, otherwise known as helper T cells, which trigger the immune response by generating cytokines and chemokines to activate other immune cells, and CD8+ T cells, known for their cytotoxic effects and role in eliminating pathogen-infected host cells [220]. Cytotoxic T cells are activated via antigen presentation on MHC class I receptors (produced by microglia and neurons). In contrast, dendritic cells, macrophages, and B cells activate T cells to amplify the immune response through presentation of antigens on MHC class II molecules [221]. In addition to response amplification, B cells also have the unique function of producing antibodies that can recognize, bind to, and neutralize a foreign substance the body has previously encountered [221]. In the brain, aberrant B cell pro- and anti-inflammatory cytokine processes are thought to contribute to multiple neuroinflammatory diseases, such as multiple sclerosis, Parkinson’s disease, and Alzheimer’s disease [222]. Notably, B cell-depleting therapies are now the standard of care for patients with neuromyelitis optica and multiple sclerosis [222,223]. 

Thus, within the brain, microglia, astrocytes, macrophages, neutrophils, dendritic cells, and B and T cells all have unique functions to stimulate and amplify the immune response and hence are important potential contributors to, and markers of, inflammation surrounding these vascular lesions. In particular, by secreting both chemokines and cytokines that together locally increase the endothelial cell surface expression of adhesion molecules for leukocyte transmigration across the vessel wall, and by stimulating BBB permeabilization, these cells together orchestrate a complex set of responses culminating in immune cell infiltration into the brain parenchyma surrounding the vascular anomalies, resulting in cerebral inflammation [224]. Notably, Winkler et al. found an increase in myeloid and lymphoid cells in bAVMs compared to control temporal lobe samples [117]. This included an increased abundance of IBA1^+^P2RY12^−^ macrophages and IBA1^+^P2RY12^+^ microglia. Myeloid cells expressed activation markers, and provocatively IBA1^+^P2RY12^−^ macrophage populations were found further away from the vasculature, suggestive of infiltration into the underlying brain parenchyma. They also found that circulating immune cells (e.g., CD8^+^ T cells) gained access to the perivascular space in bAVMs. Since the endothelium plays a key role in recruiting inflammatory cells [172], this strongly suggests that EC activation and altered vascular barrier function may contribute to the inflammatory microenvironment that is established in the nidus. Since somatic gain-of-function mutations in the endothelium have been linked to bAVM induction in zebrafish and mouse models [71], this leads us to propose that endothelial mutations may be involved in BBB compromise and subsequent inflammatory invasion. This concept is explored further below by examining inflammation and endothelial cell activity in bAVM hemorrhage and pathogenesis. 

## 5. Inflammation, bAVM Rupture, and Intracranial Hemorrhage 

Previous literature has already linked inflammatory factors to the pathogenesis and rupture of cerebral vascular malformations, particularly cerebral cavernous malformations (CCMs), with mouse models highlighting the role of gut microbiome endotoxins in CCM [225] and RNA sequencing, histology, and flow cytometry identifying neuroinflammatory astrocytes and suggesting inflammation-promoting cross-talk between astrocytes and CCM endothelium [226]. The pathogenesis of intracranial aneurysm rupture, in general, has also been associated with inflammation. Current models postulate that a hemodynamic insult precipitates inflammation, leading to matrix metalloproteinase (MMP)-mediated degradation of the extracellular matrix and programmed cell death of the predominant matrix-synthesizing cells of the vascular wall, vascular smooth muscle cells (VSMCs) [227]. These processes synergize to weaken the vessel wall, which then dilates in response to hemodynamic load, leading to aneurysm formation and eventual vessel rupture. In addition to VSMCs, macrophage recruitment and infiltration, as well as microglial and astrocyte activation in response to oxygen–glucose deprivation, play a critical role in this process by further stimulating MMP activity [120,228,229,230,231]. Astrocytes and microglia also contribute to BBB permeability through the release of pro-inflammatory factors such as VEGF-A, oncostatin M, and TNF-α [231,232,233,234,235]. Additionally, microglia are also thought to contribute to oxidative stress via production of peroxidases, iNOS, and reduced NADPH oxidase [236,237,238,239]. 

Early microarray studies analyzing gene expression determined that ruptured intracranial aneurysms samples showed increased levels of CD163, a macrophage marker, and MPO, a marker of oxidative stress and neutrophil degranulation, compared to unruptured lesions [240,241]. Additionally, genes involved in the processes of chemotaxis and cytokine signaling, as well as S100/calgranulin genes, which encode calcium-binding proteins that mediate recruitment of neutrophils and macrophages and aid in activation of RAGE (receptors for advanced glycation endproducts) and macrophage TLRs (specifically TLR4), are also elevated in ruptured samples compared to control tissue [240,241]. Notably, these lesions also feature downregulation of proteins involved in cell adhesion, such as Cadherin 5, Integrin α6 and Integrin α7, MMP-9, and Timp-3 [58,242,243,244] and structural cytoskeleton proteins such as Dystrophin and Myosin [241], and repression of anti-inflammatory transcriptional regulators, such as Krüppel-like family transcription factors (KLFs) [241,245]. Overall, the recruitment of monocyte-derived cell types, as well as the activation of resident CNS cells in the brain (e.g., microglia and astrocytes), combined with a flood of secreted proteolytic enzymes and barrier-disrupting cytokines and pro-angiogenic growth factors, together with the rise in reactive oxygen species, all act to promote vascular and basement membrane breakdown.

Similar to those with intracranial aneurysms, patients with bAVM face the life-threatening risk of lesion rupture and intracranial hemorrhage. Being able to accurately predict how the natural history and genetics of a patient place them at risk for ICH stroke compared to the risks associated with invasive therapy is a prerequisite for the effective clinical management of bAVM. However, accurate predictors of future ICH stroke in non-hemorrhagic bAVM patients have been lacking. Accordingly, identifying biomarkers or genetic variants associated with an increased risk of rupture is critical for effective case management. As numerous studies have shown that bAVM rupture, a major risk for death and disability in bAVM patients [246], is associated with specific inflammatory factors, particularly in the innate immune system [247], we will summarize the evidence linking inflammation and immune system activation to bAVM rupture. 

One of the earliest studies to document wide-spread changes in inflammatory gene expression was a targeted comparison of 42 transcripts in the blood of 20 ruptured and 20 unruptured bAVMs. The expression of inflammatory genes, such as *Fas* (TNF receptor superfamily, member 6; *FAS*), *Toll-like receptor 10* (*TLR10*), *tumor necrosis factor, alpha-induced protein 6* (*TNFAIP6*), and *IL1R1*, was elevated in ruptured bAVMs, implicating natural-killer-mediated cytotoxicity, antigen processing and presentation, T and B cell signaling, and Fc epsilon RI signaling in rupture and hemorrhage [248]. In a comparison of healthy controls, un-ruptured bAVMs and ruptured bAVMs via microarray analysis, only three genes were differentially expressed in both primary (control vs. un-ruptured) and secondary (un-ruptured vs. ruptured) analyses: *IL1RAP* (interleukin-1 receptor accessory protein), *RGL4* (Ral guanine nucleotide dissociation stimulator like 4), and *IDS* (iduronate 2 sulfatase), suggesting expression of these targets may be related to disease progression. Specifically, these genes were downregulated in unruptured vs. control specimens and upregulated in ruptured vs. unruptured specimens [248]. 

A more recent genome-wide transcriptional study of ruptured vs. unruptured bAVMs using RNA-sequencing of the resected nidus from a cohort of 65 (28 ruptured and 37 un-ruptured) patients, as well as a 35-sample validation group, determined that the top 5 out of 795 differentially upregulated genes were involved in either inflammation or oxidative stress [247]. Gene ontology analysis of these differentially expressed transcripts identified an upregulation in inflammatory-related processes, such as leukocyte recruitment, cytokine expression, and leukocyte migration, all of which contribute to damage and weakening of the vessel wall [249]. Strikingly, transcripts related to innate immune responses were also upregulated in ruptured bAVMs, which include components such as monocytes, macrophages, and phagocytic neutrophils [250]. Pathway analysis of these differentially expressed transcripts suggested that inflammation-related diseases and signaling pathways, such as Toll-like receptor (TLR) signaling, NF-κB signaling, and complement cascades were also enriched in ruptured bAVMs [247]. Conversely, genes involved in cell adhesion (DSCAM, CNTN2, CERCAM, ITGBL1, and LAMA3) and myofibril assembly (CAPN3, MYOZ1, and MYOZ2) were downregulated [247].

Critically, a previous study using MRI imaging determined that macrophage load in the nidus is correlated with a higher hemorrhage risk [251]. As a corollary, macrophages and other inflammatory cells are often observed in the vascular walls and underlying stroma of bAVMs, even in those lesions with no history of rupture [214,252,253]. Furthermore, ruptured bAVMs have a high density of M2-polarized macrophages in the perinidal dilated capillary network, further suggesting macrophages may play an important role in intracerebral hemorrhage [254]. Indeed, Wright and colleagues suggest macrophages are the dominant inflammatory cell type within bAVM, with Spetzler grade 2 (or higher) anomalies featuring higher levels of macrophages than other inflammatory cell types [255]. This aligns with studies showing increased bAVM macrophage burden in adult heterozygous null *Eng* or *Alk1* HHT mice (*Eng^−/+^*, *Alk1^−/+^*) subjected to focal angiogenic stimulation [256,257,258,259]. To further characterize macrophage dynamics in a HHT model of bAVM, Zhang and colleagues recreated *Eng* and *Alk1* mouse models from previous studies (above) and found that after eight weeks, both *Eng*- and *Alk1*-deficient mouse models had significantly greater macrophages/microglia compared to controls [260]. After observing increased macrophage burden in mice, Zhang and colleagues investigated inflammatory cells from *Eng*- and *Alk1*-deficient HHT patients using an endothelial cell and vascular smooth muscle cell co-culture to emulate an angiogenic niche. In this niche, CD34+ cells from peripheral blood more efficiently differentiated into macrophages compared to control samples, suggesting persistent infiltration and proinflammatory differentiation of monocytes may contribute to macrophage burden in bAVM [260]. Just as microglia (particularly activated microglia and DAMS) play key roles in white matter diseases where changes in cerebrovascular function often precede cognitive decline, as well as in pathologies such as stroke, these cells may also play pivotal roles in bAVM inflammation and lesion progression, although future studies are needed to test the validity of this hypothesis. 

A recent study by Winkler and colleagues [117] confirmed the presence of recruited monocytes in bAVM. Using scRNA-sequencing, they observed increased angiogenic, inflammatory and endothelial to mesenchymal transitioning (EndMT) cell populations, as well as the robust upregulation of myeloid cells featuring an activated gene signature specifically in bAVM but not control tissue samples. Moreover, CD11c^+^ dendritic cells, perivascular macrophages, IBA1+P2RY12+ microglia, and other antigen-presenting myeloid cells, as well as the infiltration of CD8^+^ T cells, were enriched in AVM tissue. They extended this analysis to profile ruptured vs. unruptured AVMs and identified an increase in a distinct infiltrating population of *GPNMB*^+^ monocytes and a decrease in smooth muscle cells in ruptured bAVMs. Ex vivo experiments suggested that these monocytes may directly lead to smooth muscle cell apoptosis via the secretion of osteopontin, a ligand for CD44, and integrin receptors on SMCs [117]. 

Whether increased macrophage recruitment or resident microglia activation leads to T and B lymphocyte recruitment or activation in bAVM remains unknown [214,261]. While it is unclear if the presence or recruitment of T and B cells is clinically significant and a consistent feature of bAVMs, the continued presence of macrophages in bAVMs may lead to unresolved inflammation, causing altered vessel remodeling and likely exacerbating the severity of the bAVM phenotype. A definitive demonstration of a detrimental role for macrophages in the etiology of bAVM, and the mechanisms underlying their recruitment to the lesions, has been lacking in the field. In this sense, blocking monocyte homing to bAVM tissue may be a possible strategy to reduce bAVM severity, although studies are needed to confirm this, including using genetic mouse models of sporadic KRAS-dependent bAVMs. 

Various cytokines and immune modulators have also been implicated in bAVM rupture. Specifically, NF-κB p65, a subunit of NF-κB, a transcription factor that regulates the expression of various pro-inflammatory genes, is upregulated in ruptured vs. unruptured bAVMs [247,262]. Additionally, recent work using transcriptional profiling studies suggest that the NF-κB pathway is enriched in ruptured bAVMs [247]. Similarly, differentially expressed genes, such as *TNFAIP6*, a factor often expressed during inflammatory processes and known to interact with serine protease inhibitors to potentially modulate neutrophil migration [263], as well as transcripts involved in ECM remodeling, cell adhesion, and cell migration [264], are upregulated in ruptured bAVM tissue [247] and peripheral blood [248], suggesting these may be biomarkers predictive of vessel rupture. Toll-like receptors, which are essential for activating the innate immune response [265], are also implicated in cerebral hemorrhage [247]. Finally, matrix metalloproteinases (MMPs), which degrade the extracellular matrix, have also been implicated in rupture. In a study comparing unruptured bAVMs to control tissue from epilepsy patients, unruptured bAVMs had increased levels of MMP-9 protein and increased activity of MMP-9 and MMP-2 (measured via gelatin zymography) [266]. Others have confirmed the increased expression of MMP-9 and MMP-2 in ruptured lesions [247]. Furthermore, IL-6, a pro-inflammatory cytokine [267], is correlated with activated MMP-9 levels, leading researchers to hypothesize that IL-6 may serve as a potential marker for hemorrhage risk [266]. 

## 6. Genetic Variants Associated with Increased Risk of bAVM Rupture

In addition to the transcriptional, imaging, and histological studies implicating inflammation in the pathophysiology of bAVMs, numerous genome-wide association studies (GWAS) and whole exome-sequencing studies (WES) have identified novel genetic variants associated with bAVM rupture. Interestingly, a genotyping study of 180 bAVM patients, 41% of which presented with ICH stroke, revealed a lack of association with genes related to angiogenesis or ICH stroke, and instead found a strong correlation with an inflammatory cytokine signature and hemorrhage [268]. Specifically, the GG genotype of the -174G>C SNP, which affects *IL-6* promoter activity [269], is a significant predictor of hemorrhagic presentation [268], and ruptured bAVMs feature increased expression of IL-6 [253]. Interestingly, *IL-6* messenger mRNA levels are strongly correlated with increased expression of *IL-1β*, *TNFα*, *IL-8*, and numerous MMPs (*MMP-3*, *MMP-9*, and *MMP-12*) [253], suggesting this cytokine may be a critical upstream regulator of lesion progression and hemorrhage. Experiments in mice revealed that stereotactic injection of recombinant IL-6 protein into the right caudate putamen (just below the cortex and adjacent to the lateral ventricle of the brain) increased expression of both *Mmp-3* and *Mmp-9*, as well as proliferation and migration of cerebral endothelial cells, further supporting a model where increased IL-6 may modulate intracerebral hemorrhage [253]. 

Polymorphisms in the promoter of *IL-1B* (specifically −511C→T and −31T→C), which encodes a cytokine that promotes leukocyte recruitment to the CNS and is associated with neuroinflammation [270], are also independent predictors of hemorrhage, further supporting the potential role of IL-1β signaling in bAVM progression [271]. Similar correlations between hemorrhage and variants in the tumor necrosis factor alpha (*TNF-α*) locus, which encodes an inflammatory cytokine that is an upstream activator of both IL-6 and MMP-9, have been observed [272].

Interestingly, the *Apolipoprotein E* (*ApoE*) genotype, which encodes for a protein involved in cholesterol metabolism, has been implicated in both ICH stroke and subarachnoid hemorrhage [273,274,275]. Two SNPs (Cys112Arg, T>C; Arg158Cys, C>T) determine the *ApoE ε2*/*ε3*/*ε4* genotype. *ApoE ε2* or *ε4* variants are known to increase ICH stroke risk in patients with cerebral amyloid angiography [276,277], and genotyping of 284 bAVM patients (18 of which presented with new ICH stroke) determined that the *ApoE ε2* genotype may influence the risk of ICH stroke in bAVM [278]. Significantly, two of these same inflammatory polymorphisms in *TNFα* (-238G>A) and *ApoE* (*ApoE ε2*) may increase vascular instability post-treatment, as they are associated with an increased risk of ICH stroke following surgical intervention [279]. In addition to inflammatory cytokines, specific polymorphisms in genes encoding extracellular membrane components, as well as adhesion molecules, such as *TNFA1P6*, have also been implicated in ruptured vs. unruptured bAVMs [247]. 

Overall, these discoveries suggest that modulating gene expression levels and/or activity may represent a viable therapeutic strategy for preventing bAVM rupture both pre- and post-surgical intervention. Furthermore, they suggest that in the future a precision medicine approach to treating patients with bAVM may be possible, and indeed beneficial, to either predict or possibly prevent adverse outcomes in bAVM, such as ICH stroke. 

### Inflammatory Genes and Biomarkers in Unruptured bAVM

In addition to the prominent role inflammation plays in bAVM hemorrhage, studies suggest dysregulated inflammation may be a key feature in early bAVM pathogenesis as *unruptured* bAVMs may also feature an inflammatory milieu. An established link between matrix metalloproteinases (MMPs) and ECM degradation in other vascular diseases [280,281,282] prompted researchers to investigate if unruptured bAVM tissue contained elevated MMPs or decreased tissue inhibitors of matrix metalloproteinase (TIMP) levels. Analysis of bAVM patient samples compared to epileptic control samples showed that bAVMs contained higher levels of total MMP-9, active MMP-9, pro-MMP-9, TIMP-1, and TIMP-3 and decreased levels of TIMP-4 [283]. Across bAVMs, those with a venous stenosis > 50% had significantly higher levels of MMP-9, leading the investigators to hypothesize that an abnormal balance of MMPs and TIMPs may contribute to bAVM instability and lesion progression [283].

Although Hashimoto and colleagues established the presence of MMP-9 in bAVM tissue, the source remained unclear, as various cell types (e.g., endothelial, vascular smooth muscle cells, neurons, astrocytes, and inflammatory cells) can produce this factor. Using an enzyme-linked immunosorbent assay [14], gelatin zymography, western blot, and immunohistochemistry, Chen and colleagues found that MMP-9 expression was restricted to inflammatory neutrophils, and that MMP-9 expression correlated with the inflammatory marker MPO and the cytokine IL-6, but not eNOS or CD31 expression, further suggesting a role for inflammation in bAVM pathogenesis [252].

Importantly, a later study of tissue from patients with unruptured and un-embolized bAVM—to control for prior hemorrhage and/or intervention acting as confounding variables—not only confirmed the colocalization of MPO and MMP-9 in bAVM tissue, but also indicated that neutrophils are the major source of MMP-9 in bAVM, and that both neutrophils (MPO^+^) and macrophages (CD68^+^) were enriched in the vascular wall, lumen, and adjacent parenchyma of unruptured bAVMs compared to control tissue [214]. 

In addition to inflammatory cells and cytokines, microarray studies showed several other inflammatory genes are upregulated in bAVM tissue. In a comparison of mRNA expression between CCM, bAVMs, and controls, Shenkar and colleagues found a 17-fold increase in the splice variant *IL8RB1*, the gene coding for the interleukin-8 receptor type B [284], a receptor for cytokine IL-8 responsible for inducing chemotaxis in target cells such as neutrophils [285]. Additionally, *ST2 interleukin-1 receptor-like 1 protein* transcripts, which encode for a receptor for IL-33 (an alarmin that notifies the immune system of tissue damage and stress [286]) are increased 10-fold in bAVMs [284]. Polymorphisms in the *IL-1B* promoter have also been associated with increased bAVM susceptibility [271]. Another pro-inflammatory receptor, TNF-related receptor (*TNFRSF10C*), was also mildly elevated (approximately three-fold) in bAVM tissue compared to controls [284]. A similar microarray study found an elevation of other inflammatory genes, such as, *TGF-β*, *angiopoietin 2*, *VEGF-A*, and *integrin αvβ3*, in bAVM tissue, further suggesting the importance of inflammation in the bAVM phenotype [287]. 

In addition to an upregulation of inflammatory genes, researchers also discovered specific mutations in inflammatory genes associated with bAVM formation. While the BMP/TGF-β superfamily has yet to be directly linked to sporadic bAVM, it is implicated in hereditary hemorrhagic telangiectasia [14], a disease known to feature systemic AVMs, including bAVMs [288]. It is well accepted that TGF-β signaling is crucial for normal vascular development [289], and mutations in *ALK1*, *ENG*, and *SMAD4*, all of which encode components of the TGF-β signaling pathway, are thought to promote disease pathogenesis by interfering with normal angiogenic signaling [290]. Co-culture models have verified the paracrine release of TGF-β by astrocytic integrin αvβ8 and demonstrated the necessity of this mechanism for endothelial differentiation [290,291]. This process is disrupted in bAVMs, with the nidus exhibiting low astrocytic integrin αvβ8 (ITGB8). Specifically, two SNPs, rs10486391 and rs11982847, in *ITGB8* have been associated with decreased astrocytic integrin αvβ8 in bAVM [290].

In addition to the TGF-β superfamily (which can control both suppressive and inflammatory immune responses), polymorphisms in other cytokines/cytokine receptors are associated with bAVM formation. For instance, Fontanella and colleagues noted SNPs in the promoter region of the *IL-1α*−889 C > T (CT or TT) and VNTR polymorphism in allele 1 of the IL-1 receptor antagonist, *IL-1RN*, in bAVM patients vs. healthy subjects [292]. MMP mutations, too, specifically the rs522616 A > G variant of MMP-3, have been associated with bAVM in a Chinese population [293]. 

Taken together, a link between inflammation and bAVM development and rupture has been established. However, many questions remain about the interaction and cross-talk between the various cell types and secreted factors in the microenvironment of the nidus. Recent single-cell genomics studies have begun to shed light on this, revealing the heterogeneity in the repertoire of inflammatory cells in lesions and how this heterogeneity is related to risk of rupture [117], but additional studies incorporating spatial transcriptomics and follow-up mechanistic studies in animal models are warranted. It is hoped that understanding the relationship between genetic variation and biomarkers with rupture risk with help to risk-stratify patients and that elucidating the mechanisms by which inflammation drives bAVM development and rupture will lead to targeted therapy to reduce the morbidity and mortality associated with this disease. 

## 7. Anti-Inflammatory Therapies for Treatment of bAVMs

Because both ruptured and unruptured bAVMs have been linked to inflammatory cells and processes, researchers have begun exploring the potential role of anti-inflammatory drugs in bAVM treatment. Treatment with minocycline, an MMP-9 inhibitor, attenuated cerebral MMP-9 activity, while pyrrolidine dithiocarbamate suppressed ~80% of cerebral MMP-9 activity, and both drugs attenuated VEGF-induced ICH stroke in a mouse bAVM model [294]. Doxycycline, a nonspecific MMP inhibitor known to suppress VEGF-induced cerebral angiogenesis, also lowered MMP-9 levels and decreased cerebro-microvessel density [295]. A small study of doxycycline in human bAVM patients (n = 10) showed decreased MMP-9 levels in treated tissue; however, this was not statistically significant, potentially due to limited sample size [296]. Anti-angiogenic drugs such as bevacizumab, a monoclonal antibody that crosses the BBB and targets VEGF-A, reduced vessel density and dysplasia in a mouse model of cerebral AVM [297]. However, anti-VEGF treatments pose significant side effects, such as increased risk of hemorrhage [14,298]. Broader-acting anti-inflammatories, such as thalidomide (and the less toxic derivative lenalidomide), which increases PDGF-b expression while decreasing CXCR4, IL-1β, and TNF-α, improved mural coverage of bAVM vessels and reduced bAVM hemorrhage in an adult *Alk1^flox/flox^* stereotactic Ad-Cre/AAV-VEGF bAVM mouse model [83]. However, like VEGF blockade, thalidomide poses its own risks, such as ischemic stroke, ischemic cardiomyopathy, and catastrophic epistaxis, each of which has previously been reported in trials testing thalidomide for treatment of HHT [14]. 

Immunosuppressive drugs, such as tacrolimus and its macroside analogue sirolimus, which act downstream of the VEGF/VEGFR2 pathway by inhibiting mTOR, have also been explored as potential therapies for bAVM [38]. In the setting of HHT, tacrolimus increased Smad1/5/8 signaling in cultured primary endothelial cells from *Alk1^−/+^* mice [38]. Normally, Alk1 phosphorylates Smad1/5/8, which, after complexing with Smad 4, translocates to the nucleus to regulate gene expression [26]. Thus, by increasing Smad signaling, these drugs may compensate for *Alk1* haploinsufficiency. While not yet specifically explored in bAVMs, work on HHT shows promise for a potentially efficacious therapy. 

Interestingly, in studies concerning allograft rejection rates, tacrolimus was shown to induce TLR4 activity in cultured murine endothelial and smooth muscle cells, increasing the production of pro-inflammatory cytokines and endothelial activation markers, and leading to vascular toxicity with chronic use [299]. Similarly, the in vitro treatment of endothelial colony-forming cells with tacrolimus impaired proliferation and migration, and upregulated pro-inflammatory factors such as TNF-α, IL6, and VCAM [300]. Thus, due to the cross-talk between endothelial activation and inflammation, tacrolimus and downstream modulators such as TLR4 inhibitors should be explored in preclinical studies as mono- or adjuvant therapeutic candidates for preventing bAVM progression. 

While disrupting mTOR signaling may prove effective for treating bAVM in HHT patients, most of these vascular anomalies arise from somatic mutations that may not affect the TGF-β signaling pathway. Considering the prevalence of activating *KRAS* variants in sporadic bAVM, MEK inhibitors are particularly attractive given their safety profile and the availability of multiple FDA-approved therapeutics. Indeed, data show that within the endothelium, constitutively active KRAS, rather than increasing PI3K and mTOR signaling, preferentially activates the MAPK-ERK pathway [40]. Furthermore, many KRAS-induced changes within endothelial cells (both in vitro and in vivo) depend upon MEK, but not PI3K, activity [71]. MEK1 kinase, a downstream target of RAF, activates ERK1/2 in the mitogen-activated protein kinase (MAPK) pathway, which plays key roles in inflammatory cell proliferation, gene expression, and apoptosis [301]. MEK 1/2 also transduce both VEGF- and histamine-induced signals, leading to increased endothelial permeability (e.g., vascular leak) [302]. Moreover, the protein kinase C (PKC)-RAF-MEK-ERK pathway increases E2F-1 gene expression (in response to native low-density lipoprotein) to regulate endothelial cell proliferation in both normal and pathologic processes such as atherosclerosis [303]. In the brain, the MAPK pathway has been linked to BBB function in rats [304], while ERK1/2 activation has been implicated in endothelial cell injury following ischemic events such as oxygen–glucose deprivation [305]. 

In terms of bAVMs, recent studies showed that treatment with a MEK-inhibitor can regress established AVM lesions and prevent hemorrhage in zebrafish expressing mutant KRAS in the embryonic endothelium [71]. Later studies in mice showed that MEK inhibition, specifically with the FDA-approved compound trametinib, prevents bAVM formation in adult mice transduced with AAV-KRAS^G12D^ [222] and normalizes cerebral angiogenesis in an endothelial-specific KRAS gain-of-function transgenic mouse model [306]. Initial results have shown some success in the clinic, as trametinib treatment of *KRAS*- or *MAP2K1*-mutated extracranial AVM resulted in fewer complications, reduced arterial inflow, and decreased AVM size [70,72]. Future studies are needed to explore the safety and efficacy of MEK inhibitor treatment for bAVM, as whether life-long MEK inhibition is a viable strategy in bAVM patients remains unclear, as is whether bAVMs will progress following withdrawal from treatment. Finally, whether MEK blockade will impact the ability of the endothelium to recruit inflammatory cells also remains unknown. Similar concerns, ranging from therapeutic resistance to long-term treatment consequences, must be balanced on a patient-by-patient basis for the use of variant-specific KRAS inhibitors [75]. Finally, we would note that despite conditional approval in 2021 by the FDA for KRAS-G12C-targeting drugs, follow-up confirmatory trial results have highlighted some potentially serious confounding factors with this therapeutic agent (see https://www.statnews.com/2023/10/03/fda-finds-potential-systemic-bias-in-amgens-kras-drug-trial-ahead-of-advisory-meeting/ (accessed on 15 October 2023) for more details and https://www.science.org/content/blog-post/confirmatory-trial-trouble-kras (accessed on 15 October 2023)).

## 8. Conclusions and Future Directions

While the use of anti-inflammatories has yet to significantly impact the clinical standard of care for most bAVM patients, mouse models and new discoveries elucidating the role of inflammation in bAVM pathogenesis and rupture suggest therapies targeting this process may represent an effective means of reducing bAVM severity and improving patient outcomes. Large randomized clinical trials are needed to further investigate the effect of existing anti-inflammatories, such as IL-1R blockade via Anakinra, on bAVMs. However, pre-clinical animal studies will likely have to be carefully thought out given the propensity of some COX-2-inhibitor-based anti-inflammatory drugs (e.g., meloxicam, Diclofenac, Celebrex, etc.) to increase the risk of gastrointestinal bleeding. Accordingly, detailed mechanistic studies in animal models are needed to identity novel, suitable targets for more selective anti-inflammatory therapeutic approaches specifically targeting the brain.

While inflammatory genes, cells, and factors are implicated in the pathogenesis and rupture of bAVM, and anti-inflammatory use exhibits potential for treatment, the cause of increased inflammatory cell infiltrate remains to be determined. We propose that alterations to the endothelium—be it via somatic mutations or damage—may play a primary role in orchestrating the migration and maintenance of such infiltrates in bAVM. This may include loss of the barrier properties of the endothelium and upregulation of adhesion molecules and other inflammatory mediators by the endothelium itself. Altered communication between ECs and surrounding perivascular cells may establish an inflammatory microenvironment that drives vascular remodeling and a proneness to rupture. Further studies are needed to test and refine this ‘endothelial centric’ view of bAVM progression and rupture and to harness this information for novel therapy. 

## Figures and Tables

**Figure 1 biomedicines-11-02876-f001:**
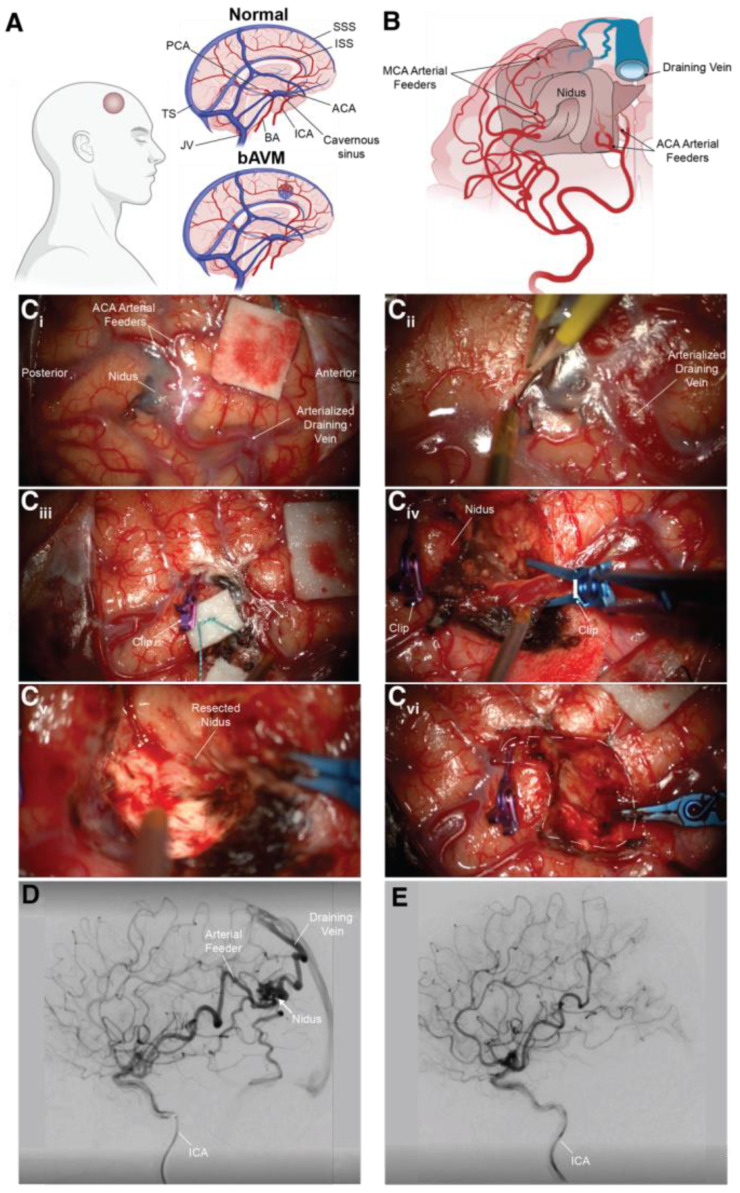
Brain arteriovenous malformations. (**A**) Schematic view of bAVM vs. normal vasculature. (**B**) Simplified schematic of unruptured right parietal bAVM in 59-year-old female with MCA and ACA feeding arteries. Surgical resection of bAVM. (**C_i_**) A parietal craniotomy being performed to gain access to the brain for the microsurgical resection of the lesion, with the feeding arterial supply from the ACA being apparent on the surface of the brain, as well as the arterialized draining vein. (**C_ii_**) Feeding arteries being are circumferentially dissected and cauterized. The primary draining vein is initially left attached. It is severed after complete arterial supply to the nidus has been cut. (**C_iii_**) ACA feeders being clipped. (**C_iv_**) Arterialized draining vein being clipped. (**C_v_**) Nidus being resected. (**C_vi_**) Parenchyma after the nidus is removed. (**D**) Angiogram (carotid artery dye injection, lateral view) of a 2 cm brain arteriovenous malformation (Spetzler–Martin grade 1), of a patient 59 years of age, before surgical intervention. (**E**) Angiogram post-resection. Abbreviations: SSS = superior sagittal sinus, ISS = inferior saggital sinus, PCA = posterior cerebral artery, TS = transverse sinus, JV = jugular vein, BA = basilar artery, ICA = internal carotid artery, ACA = anterior cerebral artery, MCA = middle cerebral artery.

**Figure 3 biomedicines-11-02876-f003:**
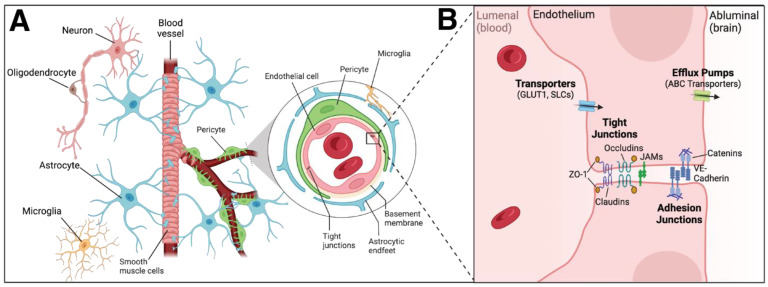
The blood–brain barrier (BBB) and neurovascular unit (NVU). (**A**) In the case of the microvasculature, the NVU is composed of endothelial cells, vascular smooth muscle cells, pericytes, and astrocytic endfeet, and serves as the building block of the BBB. (**B**) Within the endothelium, adhesion junctions, consisting of catenins and cadherins, mediate cell-to-cell adhesion, while tight junctions, consisting of occludins, claudins, JAMS 1–3, cingulin, and linker proteins of the ZO-1 family, together limit the passage of cells and molecules from the vessel lumen to the underlying brain parenchyma. This unique repertoire of junctional and adhesion proteins, along with the presence of a suite of transporters and efflux pumps, are core features of the blood–brain barrier.

**Figure 4 biomedicines-11-02876-f004:**
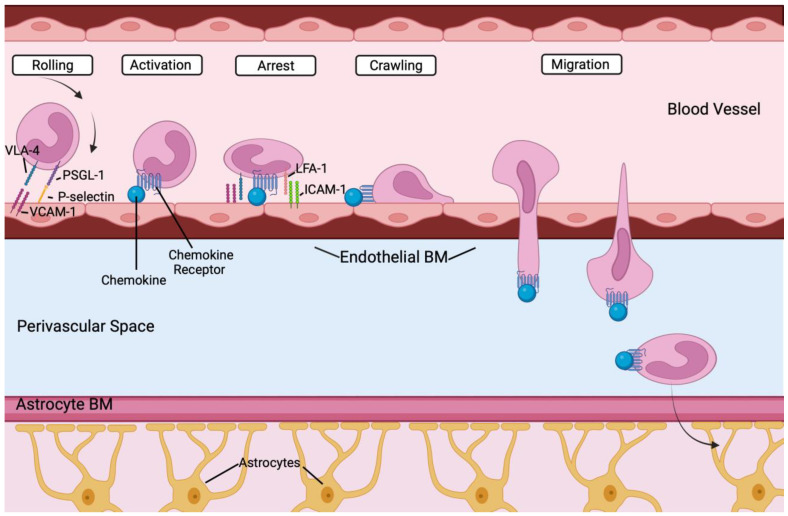
Endothelial recruitment of leukocytes to the CNS. Five main steps are used in the process of leukocyte recruitment to the CNS: (1) Rolling: the endothelium slows leukocytes through interactions between VCAM-1 and P-selectin on endothelial cells and VLA-4 and PSGL-1 on the leukocyte. (2) Activation: chemokines interact with the leukocytic chemokine receptor, activating the leukocyte. (3) Arrest: The leukocyte then upregulates VLA-4 and LFA-1 (with LFA-1 binding to endothelial ICAM-1). This allows for leukocytic attachment to the endothelial cell. (4) Crawling: the arrested, activated leukocyte crawls along the endothelium to a paracellular or transcellular pathways. (5) Migration: Chemokines in the lumen allow the leukocyte to cross the endothelium to the perivascular space. Abluminal chemokines then allow leukocytes to migrate to the CNS. Figure adapted from [172].

## Data Availability

No new data were created in the course of this study.

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
