# Peer review of "The Role and Therapeutic Implications of Inflammation in the Pathogenesis of Brain Arteriovenous Malformations"

_biomedicines, 2023, doi:10.3390/biomedicines11112876_

Round 1
Reviewer 1 Report
The authors have undertaken a comprehensive and well-organized examination of the clinical and basic science research related to brain arteriovenous malformations (bAVMs) and the potential influence of inflammatory and immune cells on both the neurovascular unit and bAVMs. Although the review articles offer a wealth of valuable information, there are areas where enhancements could be made.
1. Figure 1: Abbreviations should be included in the figure legend. The inferior sagittal sinus (ISS) is mis-labeled to SSS.
2. Overview, lines 35-36: The authors briefly mention the Richter-Suen classification for extracranial AVMs, which encompasses variables like nidus composition, the number of feeding arteries, skin involvement, among others. This classification is not used for brain AVMs; therefore, please indicate that the nomenclature of focal/multicentric AVM is part of the Richter-Suen classification for extracranial AVMs.
3. Etiology & Sequelae of brain AVMs:
lines 132 & 142: notation for heterozygous allele should be consistent: +/- vs -/+.
lines 132-135: The HHT phenotypes observed in Eng+/- and Alk+/- mice are quite subtle, rare, and variable. The statement “Notably, Eng+/- and Alk+/- adult heterozygous mutant mice also exhibit characteristics of HHT1 and 2 respectively” might not accurately convey the findings. In these heterogeneous mice, some vascular abnormalities were observed in certain organs and tissues, but the occurrence of nidus-forming AVMs or direct arteriovenous shunts was rarely observed. “Choi et al 2012” used homozygous deletion of Eng.
In HHT models, it is crucial to differentiate between well-defined arteriovenous shunts or AVMs and anomalous vasculature like dilated vessels. Furthermore, it is essential for the authors to ensure precision in citing references, associating them explicitly with particular discoveries. As an example, in the statement, "However, Eng-/+ and Alk1-/+ mouse models do develop AVMs when challenged with an inflammatory and/or pro-angiogenic stimulus, such as physical wounding or the addition of exogenous VEGF (Xu et al. 2004; Park et al. 2009; Hao et al. 2010; Mahmoud et al. 2010; Walker et al. 2011; Chen et al. 2013; Chen et al. 2014b; Choi et al. 2014; 150 Garrido-Martin et al. 2014; Han et al. 2014)," these references encompass 10 distinct studies. These studies employ varying genotyping conditions of mice, and the content of these papers do not precisely align with the statement presented.
It could be beneficial to provide a concise summary of the findings specifically related to brain AVMs. Regarding the passage in lines 154-156, which states, "specifically in the endothelium is lethal in mice, endothelial-specific deletion in adult mice does not lead to bAVMs, except in the presence of an angiogenic or inflammatory stimulus (Mahmoud et al. 2010; Garrido-Martin et al. 2014)," it's important to clarify that neither Mahmoud et al. 2010 nor Garrido-Martin et al. 2014 demonstrated the occurrence of bAVMs. The authors could consider incorporating recent reports on bAVMs in HHT models, such as those found in PMID: 4674144, 34740197, and 37219736.
lines 182-187: The authors mention the clinical management of brain AVMs, but this section requires improvement. Surgical resection offers the best cure rates, while radiosurgery is associated with reduced radiographic complete obliteration rates and long latency periods. Endovascular embolization is typically used in conjunction with surgical resection and, less commonly, with radiosurgery. Additionally, the statement in line 187, "With surgery being the only treatment for these patients," is misleading since the discussion earlier outlines multiple treatment options.
4. Inflammation and the Neurovascular Unit, lines 205-206: The sentence mentioning "lymphatic vessels that connect to cervical lymph nodes [in the brain}" is misleading. Lymph nodes are located in the neck (i.e., cervical region) and are not present in the brain.
5. Figure 2A: labels to identify the blue structures (ventricular system) and the red structure (choroid plexus) should be included. In Figure 2B/E/F, abbreviations for CVO, TJs, and CSF should be added in the figure legend.
6. Anti-inflammatory Therapies for Treatment of bAVMs:
line 806: Lebrin et al. 2010 and Boon et al. 2022 demonstrated the effects of thalidomide on AVMs other than brain AVMs, while Zhu et al. 2008 specifically showed its beneficial effects on brain AVMs. Thus, authors need to edit the statement more clearly or remove the first two references.
lines 820-824: The authors have proposed a valuable and insightful mechanism in which endothelial alterations, stemming from mutations or damage, play a pivotal role in guiding infiltrate migration and sustaining bAVM by potentially disrupting the endothelial barrier and amplifying the expression of adhesion molecules and inflammation. The inclusion of a paragraph discussing the role of endothelial cells (ECs) in the inflammatory pathogenesis of bAVM would serve to further augment the review.
Expand the treatment section to include a discussion of current studies and the mechanisms of action of immunosuppressants like tacrolimus/sirolimus. Additionally, consider discussing the potential role of PI3K and MEK inhibitors in the context of emerging drugs for the treatment of brain AVMs. Is there any evidence to suggest their implication in inflammatory circuits, in addition to their well-known implication in angiogenic pathways?
fine
Reviewer 2 Report
This topic is very interesting, just some points need to be improved:
- Lines 62-73: "Herein we discuss the epidemiology of these potentially deadly arteriovenous... or stabilizing lesions." It it not clear what is the purpose of this paper and the primary and the secondary objectives of this paper. Revise this part.
- This paper is a review. How were these papers selected?
- Lines 119-123: "Germline genetic syndromes, such as hereditary hemorrhagic telangiectasia (HHT), 119 exhibit multiple defects, such as epistaxis (nosebleeds), skin and mucosal telangiectasias 120 (dilated postcapillary venule and arterial connections), and multiple AVMs, with a pref- 121 erence for the lungs (30-50% of people with HHT) (McDonald and Stevenson 1993), liver 122 (40-70%) (McDonald and Stevenson 1993; Buscarini et al. 2004; Ianora et al. 2004), and 123 brain (10%) (McDonald and Stevenson 1993; Brinjikji et al. 2017)." This sentences is nonsense. What do the authors want to say?
- Improved discussion with there recent papers: -- doi: 10.3171/2020.10.FOCVID2071 -- doi: 10.1016/j.wneu.2023.03.116
- Lines 310-312: "in response to any pathologic insult in the brain, such as ischemic stroke or neurodegeneration, such as occurs in Alzheimer’s or Parkinson’s disease, microglia activation.. " A recent narrative review report about "New Targets and New Technologies in the Treatment of Parkinson's Disease". Revise and improved this part.
- A conclusion seciton is mandatory. What is the final message the authors want to send? What are the latest innovations?
- Some references seem old, improve them with newer ones.
- It is not clear how "4. Immune Cell Populations within the Brain" fits into this context. Anticipate the topic at the beginning of the text.
Minor editing of English language required
Round 2
Reviewer 2 Report
Good.